# The Compensatory Response of Photosystem II Photochemistry to Short-Term Insect Herbivory Is Suppressed Under Water Deficit

**DOI:** 10.3390/insects16090984

**Published:** 2025-09-21

**Authors:** Julietta Moustaka, Ilektra Sperdouli, Stefanos S. Andreadis, Nikoletta Stoikou, Kleoniki Giannousi, Catherine Dendrinou-Samara, Michael Moustakas

**Affiliations:** 1Department of Botany, Aristotle University of Thessaloniki, 54124 Thessaloniki, Greece; ioumoustaka@gmail.com; 2Institute of Plant Breeding and Genetic Resources, Hellenic Agricultural Organisation-Demeter (ELGO-Demeter), 57001 Thessaloniki, Greece; esperdouli@elgo.gr (I.S.); sandreadis@elgo.gr (S.S.A.); stoikounikoletta@gmail.com (N.S.); 3Laboratory of Inorganic Chemistry, Department of Chemistry, Aristotle University of Thessaloniki, 54124 Thessaloniki, Greece; klegia@chem.auth.gr (K.G.); samkat@chem.auth.gr (C.D.-S.)

**Keywords:** *Tuta absoluta*, *Solanum lycopersicum*, herbivory, drought stress, non-photochemical quenching (NPQ), effective quantum yield of PSII (Φ*_PSII_*), electron transport rate (ETR), excitation pressure, compensatory photosynthesis, hormesis

## Abstract

The tomato leafminer (*Tuta absoluta*) can cause extensive damage to tomato plants (*Solanum lycopersicum*). After 20 min of feeding by *T. absoluta* larvae, a differential response mechanism of photosystem II (PSII) was observed in well-watered and mildly drought-stressed plants. In well-watered plants, whole-leaf PSII photochemistry compensated for the decreased PSII photochemistry at the feeding zone by increasing its light energy use efficiency. In contrast, in mildly drought-stressed plants, this compensatory capability was suppressed. “Plant health status” seems to be critical in the response of PSII photochemistry to insect herbivory. It is concluded that, as a result of increased drought stress episodes, increased crop damage by insect herbivory is likely to occur.

## 1. Introduction

Sustaining and improving crop photosynthesis under abiotic and biotic stresses is critical for agricultural production [1]. Under such adverse environmental conditions, plants experience a sequence of alterations at biochemical and molecular levels [2]. Such deviations are more noticeable under herbivory and drought, two major environmental stressors that can trigger functional consequences to crops [1,3]. However, the combined effects of drought and herbivory do not always result in a simple additive response, but rather, they may elicit complex and interconnected physiological and molecular mechanisms [2,4]. Drought-sensitive crops like tomato (*Solanum lycopersicum* L.), which is the most extensively grown vegetable worldwide [5], are particularly susceptible to diminished water availability as a result of climate change [6,7,8].

Plants are subjected to drought stress when soil moisture becomes insufficient or when transpiration rates are elevated [9,10,11,12]. This water scarcity disrupts the plant’s osmotic balance, hinders photosynthetic efficiency, impairs growth, and affects the overall energy homeostasis, ultimately leading to reduced agricultural productivity [13,14,15]. Among the various environmental challenges linked to climate change, drought stands out as a major limiting factor for global crop yield [16,17]. Additionally, climate change has significant implications for plant–herbivore dynamics. Elevated temperatures, increased atmospheric CO_2_ concentrations, and water stress can intensify herbivore feeding on plants and influence insect development patterns [3]. Combining these issues, the growing resistance of pests to chemical control agents poses a further threat to sustainable crop production [18,19,20,21].

Drought stress conditions can modify plant metabolism and defense reactions, resulting in changes in the synthesis and emission of volatile organic compounds [22,23,24]. During drought stress, tomato suppresses the emission of specific herbivore-induced plant volatiles (HIPVs) that enhance the defense responses against herbivores [25]. Likewise, water limitation in *Vicia faba* alters volatile organic compounds (VOCs) emission, increasing attractiveness to parasitoids [26]. Nevertheless, these reactions can differ significantly between plant species and insects [27], as revealed in *Brassica oleracea* L., where drought-stressed plants attracted more moths for oviposition, while the behavior of other parasitoids was not affected [28]. Drought stress decreases photosynthetic efficiency and changes the biosynthesis of defense-related compounds [29,30], compromising plants’ ability to defend against herbivores and to attract natural enemies [27,31,32,33,34]. Still, herbivorous insects have been reported to perform better on drought-stressed host plants [34]. However, drought stress can induce the synthesis of stress-specific volatile organic compounds, such as those connected with oxidative stress and stress signal responses to insect herbivory [35,36].

Plants utilize both constitutive and inducible defense mechanisms, along with additional response strategies, to counter herbivore attacks [37,38]. In response to these, herbivores have developed specific adaptations that enable them to bypass these plant defenses [37,38]. The interaction between plants and insect herbivores is regulated by complex molecular pathways, which also influence subsequent compensatory mechanisms in the plant [39,40]. Gaining insight into the broad spectrum of plant responses to herbivory requires us to analyze how physical damage affects physiological processes, particularly photosynthesis [38,41,42,43]. Reports on leaf-level photosynthetic responses following herbivory vary widely, including increases, no observable changes, or reductions in photosynthetic capacity [42,44,45].

Conventional methods for estimating productivity loss resulting from herbivory in agriculture often fail to account for the impact on photosynthesis in the remaining intact leaf tissue [38,46]. A detailed analysis of the photosynthetic response allows for a more accurate understanding of how undamaged areas are physiologically affected [38]. Chlorophyll fluorescence has been widely used to assess the functionality of the photosynthetic apparatus and to evaluate plant tolerance to both biotic and abiotic stresses [15,39,45,47,48]. Since biotic and abiotic stresses alter light energy utilization, chlorophyll fluorescence is effective in detecting associated impairments in photosynthesis, as it is a noninvasive, cheap, and accurate method [7,8,38,39,45,49,50]. However, photosynthetic activity is spatially heterogeneous across the leaf surface, limiting the standard point-based chlorophyll fluorescence measurements [12,47,51,52,53,54]. The advent of chlorophyll fluorescence imaging has addressed this limitation by enabling spatially resolved analysis of photosynthetic heterogeneity at the whole-leaf level [12,54,55]. Insect herbivory, along with other biotic stresses, is known to alter photosynthetic activity, most often reported to reducing it, although compensatory responses can also occur [38,42,44,45,49,50]. Photosynthesis consists of two main components: the electron transport chain and carbon fixation [56]. Since the light reactions of photosynthesis offer the energy required for the synthesis of compounds used in plant defense, they have to be incorporated into the plant’s reaction to herbivory [57]. Photosystem II (PSII) is a multi-subunit complex that converts solar energy to chemical energy and, by oxidizing water, produces the atmosphere’s oxygen, thus sustaining life on Earth [15,58,59,60]. PSII is considered to be vulnerable to both biotic and abiotic stress conditions [61,62].

*Tuta absoluta* (Meyrick) (Lepidoptera: Gelechiidae) is characterized as one of the most damaging phytophagous pests of tomato production worldwide, causing serious problems in both greenhouse conditions and open fields by diminishing crop yield [63,64,65]. Although plant–insect interactions have been broadly studied, limited investigations have been conducted on how the photosynthetic mechanism responds to insect herbivore feeding [49,57]. Yet, the response mechanism of PSII photochemistry in well-watered compared to drought-stressed plants to insect herbivory remains to be elucidated. Here, we studied how mild drought stress influences the response of the photosynthetic mechanism of the most extensively grown vegetable worldwide to short-term insect herbivore feeding of the most damaging phytophagous pest of tomato. Our aim was to assess if global climate change, with the associated increased drought stress episodes, is going to influence plant–insect interactions and increase the crop damage by insect herbivory.

## 2. Materials and Methods

### 2.1. Plant Material and Growth Conditions

Tomato (*Solanum lycopersicum* L. cv Ecstasy) plants grew in 5 L round plastic pots that contained peat moss (Terrahum, Fytoprostasia, Chalkida, Greece) and perlite (Geoflor, Perlite Hellas, Volos, Greece) (1:1 *v*/*v*) in an insect-proof greenhouse, with a 23 ± 1/20 ± 1 °C (day/night) temperature, a 14 h photoperiod with photosynthetic a photon flux density (PPFD) of 250 ± 10 μmol quanta m^−2^ s^−1^, and a day/night relative humidity of 70 ± 5/80 ± 5%.

### 2.2. Tuta Absoluta

*Tuta absoluta* (Meyrick) (Lepidoptera: Gelechiidae) individuals used in this experiment originated from a colony maintained in the Entomology Lab of the Institute of Plant Breeding and Genetic Resources, Hellenic Agricultural Organization—Demeter (Thermi, Greece), as described previously [49]. Approximately 100 adults of *T. absoluta* were transferred with a simple mouth-operated aspirator (BioQuip Products, Compton, CA, USA) into pop-up breeding cages (40 × 40 × 60 cm) with a single vinyl window and zip closure (Raising Butterflies, Salt Lake City, UT, USA), containing three-to-four insect-free tomato plants, and were allowed to oviposit for 24–48 h [49]. After being allowed to oviposit for 24–48 h, the infested leaves were cautiously detached and positioned inside smaller breeding cages (30 × 30 × 30 cm) to permit larval development to the third instar. Third-instar larvae (L_3_) used in the study were starved for 24–30 h prior the experiments.

### 2.3. Experimental Design

Tomatoes used for the experiments were at the flowering growth stage 61 according to the BBCH numerical scale, a system for the uniform coding of growth stages [66]. The leaflet, of a well-watered or mildly drought-stressed plant, was enclosed in the measurement chamber of the fluorometer and the photosynthetic efficiency was measured before herbivore feeding. Subsequently, one third-instar larva (L3) was added in a leaflet of either a well-watered or mildly drought-stressed plant, and a cap was placed over to act as an enclosure [38]. After 20 min of feeding, the larva was removed, and the PSII function was measured immediately.

### 2.4. Water Stress Treatments

Tomato plants in the greenhouse were either adequately watered every 2 days and considered well-watered (WW) plants (control plants) or mildly drought-stressed (MDS) plants that were not watered by withholding the irrigation until the soil water content reduced to almost half that of the control plants.

### 2.5. Soil and Leaf Water Water Content Determination

The soil volumetric water content (SWC) was measured with a ProCheck device connected to the 5TE sensor (Decagon Devices, Pullman, WA, USA) [67]. Leaf water content of tomato leaflets was evaluated using the electronic moisture balance (MOC120H, Shimadzu, Tokyo, Japan) [68].

### 2.6. Chlorophyll Fluorescence Analysis

Chlorophyll fluorescence measurements were performed with the Imaging-PAM Fluorometer M-Series MINI-Version (Heinz Walz GmbH, Effeltrich, Germany), as described in detail previously [69]. All leaves were dark-adapted for 20 min before measuring F*o* and F*m* (the minimum and maximum, respectively, chlorophyll *a* fluorescence). F*m*′ (maximum chlorophyll *a* fluorescence in the light) was acquired with saturating pulses (SPs) every 20 s for 5 min after application of actinic light (AL), while F*o*′ (minimum chlorophyll *a* fluorescence in the light) was computed as F*o*′ = F*o*/(F*v*/F*m* + F*o*/F*m*′) [70]. Steady-state photosynthesis (F*s*) was measured after 5 min of illumination time with the AL of 250 μmol photons m^−2^ s^−1^, corresponding to the growth light intensity (GLI), and with 1000 μmol photons m^−2^ s^−1^ AL, as a high light intensity (HLI). The chlorophyll fluorescence parameters, which are described in detail in Appendix A, were estimated by using the software Win V2.41a (Heinz Walz GmbH, Effeltrich, Germany).

### 2.7. Statistics

Statistical analysis was performed with R software (version 4.3.1, R Core Team, 2023). Statistically significant differences were evaluated by two-way ANOVA for each parameter, with treatments (well-watered or mildly drought-stressed) and insect herbivory (before and after feeding) as factors, followed by Tukey’s post hoc test. The data were checked for normality and homogeneity of variance with the Shapiro–Wilk test and Levene’s test, respectively. Values were considered significantly different at *p* < 0.05. A linear regression analysis was also performed.

## 3. Results

### 3.1. Leaf Water Content and Soil Water Content

The soil water content of mildly drought-stressed (MDS) tomato plants decreased by 57% compared to well-watered (WW, control) plants. However, despite this high decrease in soil water content, the leaf water content decreased by only 3% (Table 1).

### 3.2. Impact of Herbivore Feeding on the Maximum Efficiency of Photosystem II Photochemistry and the Efficiency of the the Oxygen-Evolving Complex

The maximum efficiency of PSII photochemistry (F*v/*F*m*) in WW tomatoes decreased after 20 min of larvae feeding (Figure 1a), but MDS did not result in any significant difference in the F*v/*F*m* ratio compared to before feeding, or to 20 min of larvae feeding in well-watered plants (Figure 1a). The response pattern of the efficiency of the oxygen-evolving complex (F*v/*F*o*) (Figure 1b) was similar to that of the F*v/*F*m* ratio (Figure 1a), suggesting a close correlation between them.

### 3.3. Impact of Herbivore Feeding on Light Energy Use Efficiency

The allocation of absorbed light energy at PSII, for (i) beneficial use for photochemistry (Φ*_PSII_*), (ii) heat dissipation loss (Φ*_NPQ_*), or (iii) loss by a nonregulatory way (Φ*_NO_*) [71], was measured before and after 20 min of larvae feeding in WW or MDS tomatoes. The effective quantum yield of PSII photochemistry (Φ*_PSII_*) at both the growth light intensity (GLI, 250 μmol photons m^−2^ s^−1^) and the high light intensity (HLI, 1000 μmol photons m^−2^ s^−1^) increased after 20 min of larvae feeding in the WW tomatoes (Figure 2a,b). At MDS, Φ*_PSII_* decreased compared to controls, but larvae feeding had no effect compared to drought stressed tomatoes before feeding (Figure 2a,b).

Τhe quantum yield of regulated non-photochemical energy loss in PSII (Φ*_NPQ_*) decreased after 20 min of larvae feeding in well-watered tomatoes, at both 250 (GLI)- and 1000 (HLI) μmol photons m^−2^ s^−1^ light intensity (Figure 3a,b). Twenty min of larvae feeding in MDS tomatoes did not result in any significant difference in Φ*_NPQ_* at the HLI, compared to before feeding or to that of 20 min feeding in WW plants (Figure 3b), while at the GLI Φ*_NPQ_* decreased further by larvae feeding in MDS tomatoes, compared to before feeding (Figure 3a).

The quantum yield of non-regulated energy loss in PSII (Φ*_NO_*) at the GLI increased after larvae feeding, compared to controls (Figure 4a), but remained at the level of the controls at HLI (Figure 4b). Φ*_NO_* increased by MDS at both light intensities (Figure 4a,b), but Φ*_NO_* after larvae feeding in MDS tomatoes increased further only at the GLI (Figure 4a). At the HLI Φ*_NO_* after larvae feeding in MDS tomatoes was maintained at the pre-feeding level (Figure 4b). Thus, larvae feeding increased Φ*_NO_* in both WW and MDS tomatoes only under the low light intensity that corresponded to the GLI.

### 3.4. Impact of Herbivore Feeding on the Electron Transport Rate and the Photoprotective Mechanism

The electron transport rate (ETR), at both the GLI and the HLI, increased after 20 min of larvae feeding in WW tomatoes, while the ETR decreased in the MDS tomatoes compared to controls. However, larvae feeding on MDS tomatoes had no effect, at both the GLI and the HLI (Figure 5a,b).

The photoprotective heat dissipation by the non-photochemical quenching (NPQ) mechanism decreased at the GLI after 20 min of larvae feeding in WW tomatoes (Figure 6a), but at the HLI it was unaffected (Figure 6b). In drought-stressed tomatoes NPQ decreased compared to controls at both GLI and HLI (Figure 6a,b), while larvae feeding decreased NPQ further at the GLI (Figure 6a), but at the HLI it remained the same as before feeding (Figure 6b).

### 3.5. The Fraction of Open PSII Reaction Centers and Their Efficiency Before and After Herbivore Feeding

The fraction of open PSII rection centers (q*p*), at both the GLI and the HLI, increased after 20 min of larvae feeding in the WW tomatoes. In MDS tomatoes, q*p* decreased, compared to controls, but larvae feeding had no effect, compared to drought-stressed tomatoes before feeding, at both the GLI and HLI (Figure 7a,b).

The efficiency of the excitation energy capture by the open PSII reaction centers (F*v*′/F*m*′), at both the GLI and the HLI, did not differ between the treatments, showing a remarkable resistance to both biotic and abiotic stressors (Figure 8a,b).

### 3.6. The Excess Excitation Energy and the Excitation Pressure at PSII Before and After Herbivore Feeding

The excess excitation energy (EXC), at both the GLI and the HLI, decreased after 20 min of larvae feeding in well-watered tomatoes, while drought stress increased EXC compared to controls. The larvae feeding had no effect on EXC, compared to drought-stressed tomatoes before feeding, at both the GLI and the HLI (Figure 9a,b).

The excitation pressure at PSII (1-q*L*) did not change after 20 min of larvae feeding in well-watered tomatoes at the GLI (Figure 10a), while at the HLI in well-watered tomatoes, it decreased after 20 min of larvae feeding (Figure 10b). Drought stress increased the excitation pressure at PSII at both the GLI and the HLI, compared to the well-watered tomatoes (Figure 10a,b). However, in MDS tomatoes after larvae feeding, the excitation pressure (1-q*L*) increased further only at 250 μmol photons m^−2^ s^−1^ light intensity (Figure 10a), while at 1000 μmol photons m^−2^ s^−1^ light intensity, it remained at pre-feeding level (Figure 10b).

### 3.7. Correlation of the Efficiency of the Oxygen-Evolving Complex with the Maximum Efficiency of PSII

The patterns of the maximum efficiency of photosystem II photochemistry (F*v/*F*m*) and of the efficiency of the oxygen-evolving complex (F*v/*F*o*) were closely related (Figure 1a,b), as it was documented by the significant positive correlation that they showed in the regression analysis (R^2^ = 0.974, *p* < 0.001) (Figure 11).

### 3.8. Correlation of the Excess Excitation Energy with the Effective Quantum Yield of PSII

A significant negative correlation was found between the excess excitation energy at PSII (EXC) and the quantum yield of PSII photochemistry (Φ*_PSII_*) (Figure 12a,b). A decreased Φ*_PSII_* was corelated with an increased excess excitation energy at both the GLI (R^2^ = 0.9492, *p* < 0.001, Figure 12a) and the HLI (R^2^ = 0.9906, *p* < 0.001, Figure 12b).

### 3.9. Correlation of the Open PSII Reaction Centers with the Effective Quantum Yield of PSII

A significant positive correlation between the redox state of quinone A (Q_A_), representing the portion of the open PSII reaction centers (q*p*), and the effective quantum yield of PSII photochemistry (Φ*_PSII_*), at both the GLI (R^2^ = 0.8662, *p* < 0.001, Figure 13a) and the HLI (R^2^ = 0.9746, *p* < 0.001, Figure 13b), was discovered, suggesting that the efficiency of the excitation energy capture by the open PSII reaction centers (F*v*′/F*m*′) was not significantly influenced by the treatments and had a negligible impact on the increase or decrease in the quantum yield of PSII photochemistry.

### 3.10. The Spatiotemporal Heterogeneity of PSII Function Before and After Herbivore Feeding

Representative color pictures of the whole area of the tomato leaves for the parameters Φ*_PSII_*, Φ*_NPQ_*, Φ*_NO_*, and q*p* (captured at the growth light intensity, GLI) of WW and MDS tomatoes, before and after 20 min of larvae feeding, are shown in Figure 14. After 20 min of larvae feeding, in WW tomatoes the effective quantum yield of PSII photochemistry (Φ*_PSII_*) increased at the whole-leaf level despite the decreased Φ*_PSII_* at the feeding areas (shown by white arrows in Figure 14). At the same time, the significantly decreased quantum yield of heat dissipation (Φ*_NPQ_*) resulted in increased values of the quantum yield of non-regulated energy loss in PSII (Φ*_NO_*), compared to before feeding. In MDS tomatoes, it decreased.

Φ*_PSII_* and Φ*_NPQ_* caused a higher increase in Φ*_NO_*. Larvae feeding in MDS tomatoes reduced further both the Φ*_PSII_* and Φ*_NPQ_*, triggering a higher increase in Φ*_NO_*.

## 4. Discussion

Crops are exposed to a range of abiotic and biotic stresses, either separately or simultaneously, which can vary significantly across space and time [72,73]. In response to these stressors, plants undergo various metabolic alterations that are particularly pronounced under conditions of drought and herbivory—two predominant abiotic and biotic stressors—that can lead to functional impacts such as diminished plant vigor and yield [3,4,73,74,75]. However, plants can react to the disruption of their homeostasis, induced by any abiotic or biotic stress factor, by exhibiting adaptive responses, often resulting in a compensatory reaction characterized as hormesis [59,73,76,77,78,79]. The term “hormesis” denotes an overcompensatory physiological adjustment following a disruption of homeostasis, a phenomenon that has been observed across a wide range of organisms, irrespective of the nature of the stressor or the specific metabolic pathways involved [59,76,77,78,79,80,81,82,83,84,85,86,87,88,89].

Understanding the molecular mechanisms that initiate hormesis in plants is key to enhancing crop productivity [87,90]. The effectiveness of hormesis largely depends on factors such as the selected dose range, exposure duration, and the experimental setup [59,80,82,85,90,91,92]. Twenty min of feeding induced an enhancement of PSII photochemistry in WW plants resembling a hormetic response, which was not observed after feeding in MDS tomatoes. It is well recognized that PSII hormetic responses are evident only when studies are carefully and appropriately designed [59,90]. Therefore, the effect of longer-duration feeding may not have the same response.

The effective quantum yield of PSII photochemistry (Φ*_PSII_*) (Figure 2a,b), and, respectively, the electron transport rate (ETR) (Figure 5a,b), of the whole leaf in WW plants after 20 min of larvae feeding overcompensated for the decreased Φ*_PSII_* observed at the feeding area, compared to the pre-feeding whole-leaf level (Figure 14). This increased Φ*_PSII_* at both the GLI and the HLI after herbivore feeding in WW plants seems to be a hormetic response to the disruption of homeostasis. In contrast, Φ*_PSII_* and ETR in MDS plants after 20 min of larvae feeding did not compensate at the whole-leaf level for the reduced Φ*_PSII_* because of drought stress (Figure 14). The increased Φ*_PSII_* and ETR at the whole-leaf level in WW plants after 20 min of larvae feeding was the result of an increased fraction of open PSII reaction centers (q*p*) (Figure 7a,b), since there was no difference in the efficiency of the open PSII reaction centers (F*v*′/F*m*′) before and after feeding (Figure 8a,b). Plants have to accomplish defense strategies to adjust their PSII function when they experience herbivore attacks [38].

The decreased maximum efficiency of PSII photochemistry (F*v/*F*m*) after 20 min of larvae feeding on WW plants suggests possible photoinhibition. Photoinhibition appears when the light energy absorbed by the leaves is neither used for photosynthesis nor dissipated harmlessly as heat by the photoprotective mechanism of non-photochemical quenching (NPQ), or alternatively, when the primary cause is the excitation of Mn in the oxygen-evolving complex (OEC) by photons [93,94,95,96,97,98,99]. Thus, photoinhibition can appear by dual mechanisms [95,97,100]. In our experiments the observed photoinhibition after 20 min of larvae feeding on WW plants led to a reduced efficiency of the OEC, as estimated from the ratio F*v/*F*o* [101,102,103,104,105,106,107]. The superior maximum efficiency of PSII photochemistry (F*v/*F*m*) in the WW leaves (Figure 1a) was in agreement with the parallel response pattern of the OEC in the same leaves (Figure 1b). It is well documented that a lower OEC efficiency also indicates a decrease in F*v*/F*m* [108,109]. Consistent with this, a significant positive correlation between the OEC efficiency and the maximum efficiency of PSII photochemistry (F*v/*F*m*) was revealed by regression analysis (Figure 11). Photoinhibition has been shown to be correlated with reduced efficiency of the OEC [110,111,112,113] and is frequently connected with crop production [114].

Despite the decrease in the F*v/*F*m* ratio after 20 min of larvae feeding on WW plants (Figure 1a), the effective quantum yield of PSII photochemistry (Φ*_PSII_*), at both the GLI and the HLI, increased after larvae feeding in the WW tomatoes (Figure 2a,b), thus corroborating the view that the observed photoinhibition was not related to reduced light energy use efficiency, but to reduced efficiency of the OEC [110,111,112,113]. Opposing our results, Hu et al. [1] recently reported that herbivory-induced PSII photoinhibition in tomatoes, observed by a reduction in F*v*/F*m*, was associated not only with a decrease in non-photochemical quenching (NPQ) but also with a decrease in the effective quantum yield of PSII photochemistry (Φ*_PSII_*) after larval feeding. Contradictory results likely stem from the diversity of experimental approaches, embracing diverse time-point measurements [46,115,116], and/or to the absence of spatiotemporal measurements [38,39,117,118]. Also, the plant species examined (e.g., potato, tomato) or the insect tested (e.g., *Tuta absoluta, Spodoptera exigua*) can lead to different conclusions [38,49].

To prevent photoinhibition, primarily of PSII, though PSI can also be affected to a lesser extent, plants have evolved regulatory mechanisms to manage excess excitation energy [119,120,121,122]. The increased fraction of open PSII reaction centers after 20 min of larvae feeding on WW plants (Figure 7a,b) that caused an increased Φ*_PSII_* resulted in decreased excess excitation energy (EXC) at the whole-leaf level (Figure 9a,b). This was also documented by the relationship between the excess excitation energy (EXC) and the effective quantum yield of PSII photochemistry (Φ*_PSII_*) (Figure 12a,b). On the other hand, the increased excess excitation energy in the MDS leaves, before and after larvae feeding on these leaves (Figure 9a,b), was attributed to the fact that the absorbed light energy was not used for photochemistry (Figure 2a,b), nor it was safely dissipated by the NPQ process as heat (Figure 6a,b). “Excess excitation energy” that is reaching the PSII and is not used for photochemistry must be dissipated non-radiatively as heat (NPQ) to safeguard the photosynthetic apparatus [123,124,125] and avoid any photodamage [126,127]. Photodamage of the photosynthetic apparatus [128] can lead to photoinactivation [129,130], decreasing photosynthetic activity and causing photoinhibition [131,132].

Under most environmental stress conditions, the amount of light energy absorbed by PSII and PSI exceeds the capacity for photochemical utilization, leading to elevated production of reactive oxygen species (ROS), including singlet oxygen (^1^O_2_), superoxide anion radicals (O_2_^•−^), and hydrogen peroxide (H_2_O_2_) [125,133,134,135,136,137]. When the amount of absorbed light energy exceeds the amount that can be used for photochemistry, then PSII excitation pressure increase [132]. Excitation pressure is defined as the relative measure of the reduction state of Q_A_, representing the percentage of closed PSII reaction centers. In our findings, 20 min of larvae feeding decreased the excitation pressure at PSII (1-q*L*) at the HLI in WW tomatoes, suggesting that a molecular mechanism for lessening the biotic stress effects may exist under HLI and herbivore feeding.

The small decline in leaf water content, despite the high decrease in soil water content (Table 1), suggests an osmotic adjustment [138,139,140]. One mechanism of osmotic adjustment is the accumulation of compatible solutes, or compatible osmolytes [138,139]. Under drought stress, the induction of proline, soluble sugars, and anthocyanin synthesis contributes to osmotic adjustment, helping plant cells acclimate to water stress by retaining the water content [138,139]. However, despite the small decline in leaf water content during MDSm, the effective quantum yield of PSII photochemistry (Φ*_PSII_*) decreased (Figure 2a,b). This decrease in Φ*_PSII_* could not be balanced by Φ*_NPQ_* (Figure 3a,b), and, as a result, Φ*_NO_* increased (Figure 4a,b).

Larvae feeding in WW tomatoes induced a more oxidized redox state of quinone A (Q_A_), representing a higher fraction of open PSII reaction centers (q*p*), and enhanced PSII photochemistry, compared to before feeding. In contrast, MDS and larvae feeding in MDS tomatoes triggered a decrease in the fraction of open PSII reaction centers (q*p*) and reduced PSII photochemistry, compared to WW tomatoes. Drought stress and herbivore feeding resulted in an increased quantum yield of non-regulated energy loss in PSII (Φ*_NO_*) at the GLI, while at HLI Φ*_NO_* increased only by drought stress (Figure 4a,b). Φ*_NO_* comprises chlorophyll fluorescence internal conversions and intersystem crossing, causing ^1^O_2_ generation through the triplet state of chlorophyll (^3^chl*) [8,123,141,142]. Thus, at HLI, drought stress results in a higher photodamage risk than does the insect herbivory. This drought-stress-induced photodamage is related to the rise in ROS production [10,143]. It can be suggested that the molecular mechanisms for lessening biotic stress effects, such as decreasing ROS generation, may exist under herbivore feeding at HLI. On the other hand, the quantum yield of regulated energy dissipation (Φ*_NPQ_*) seems to have no role in counteracting the rise in ROS production under herbivore feeding at HLI (Figure 3b). Herbivore feeding has been reported to induce a compensatory photosynthetic response [38,39,144,145]. The effects of drought stress on herbivore survival have been reported to range from strongly positive (50% increase) to strongly negative (80% decrease), while they were found to be extremely different among closely related plant species and between co-occurring insect herbivores [146]. Plant responses can be unusually specific to the herbivore species or even to the age-related stage of the herbivore [147,148].

The spatial leaf heterogeneity that was observed under all treatments, in all chlorophyll fluorescence parameters, demonstrates the advantage of using chlorophyll fluorescence imaging technique instead of the standard point-based chlorophyll fluorescence measurements [12,15,54].

## 5. Conclusions

In brief, our results show that in well-watered (WW) plants, photosynthetic efficiency was only locally suppressed at the feeding area of *Tuta absoluta*, while at the whole-leaf level photosynthetic efficiency improved, indicating a compensatory response. In contrast, in mildly drought-stressed (MDS) plants, photosynthetic efficiency decreased at both the feeding area and the whole leaf after 20 min of larvae feeding. “Plant health status” seems to be critical in the response of PSII photochemistry to insect herbivory. Since drought stress suppresses PSII compensatory response to insect herbivory, it seems that the increased duration, frequency, and intensity of drought stress episodes due to climate change will likely result in higher yield loss attributed to the higher susceptibility of tomatoes to insect pests.

The enhancement of PSII photochemistry under short-term herbivore feeding in the WW plants seems to be a hormetic response to the disruption of homeostasis triggered by defense strategies that adjust PSII function in response to herbivore attack. In contrast, under MDS these defense strategies could not be activated in response to herbivory. Therefore, under projected climate change scenarios, with the increased drought stress episodes which are estimated to appear, and thus to influence plant–insect interactions, a decreased photosynthetic efficiency is likely to occur, attributed to the interaction of drought stress and insect herbivory. However, based on our findings, drought stress results in higher photodamage risk than does insect herbivory, so future research breeding programs must focus on developing drought-resilient crops rather than herbivore-resistant crops.

Future research should explore PSII responses under longer durations of herbivore feeding in WW and MDS crops in field experiments to uncover any similarities in PSII responses with the present study.

## Figures and Tables

**Figure 1 insects-16-00984-f001:**
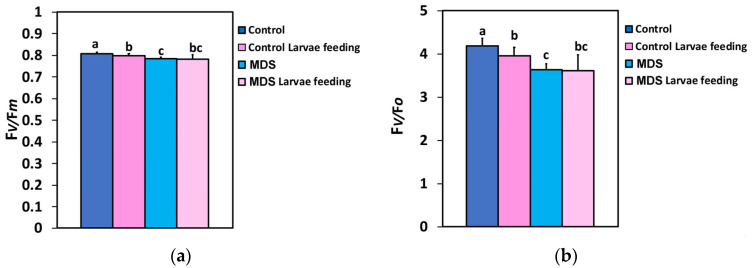
The maximum efficiency of PSII photochemistry (F*v/*F*m*) (**a**), and the efficiency of the oxygen-evolving complex (F*v/*F*o*) (**b**) before and after 20 min of larvae feeding, in well-watered (WW, control) or mildly drought-stressed (MDS) tomatoes. Standard deviations (SD) are shown by bars. Different lower-case letters reveal significant differences at *p* < 0.05 (*n* = 6).

**Figure 2 insects-16-00984-f002:**
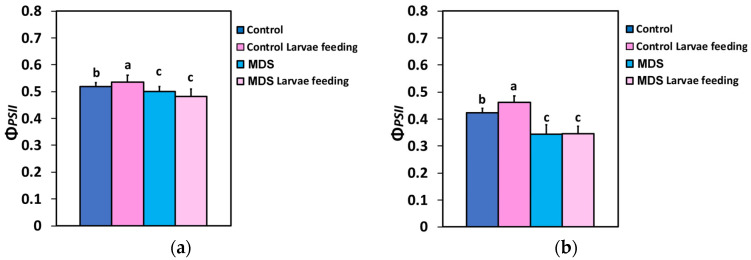
The effective quantum yield of PSII photochemistry (Φ*_PSII_*) at actinic light (AL) of 250 μmol photons m^−2^ s^−1^, corresponding to the growth light intensity (GLI) (**a**), and at actinic light (AL) of 1000 μmol photons m^−2^ s^−1^, corresponding to a high light intensity (HLI) (**b**), measured before and after 20 min of larvae feeding, in well-watered (WW, control) or mildly drought-stressed (MDS) tomatoes. Standard deviations (SD) are shown by bars. Different lower-case letters reveal significant differences at *p* < 0.05 (*n* = 6).

**Figure 3 insects-16-00984-f003:**
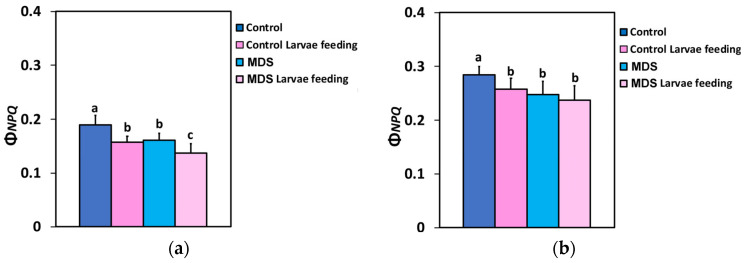
Τhe quantum yield of regulated non-photochemical energy loss in PSII (Φ*_NPQ_*) at 250 μmol photons m^−2^ s^−1^ AL (**a**) and at 1000 μmol photons m^−2^ s^−1^ AL (**b**), before and after 20 min of larvae feeding, in well-watered (WW, control) or mildly drought-stressed (MDS) tomatoes. Standard deviations (SD) are shown by bars. Different lower-case letters reveal significant differences at *p* < 0.05 (*n* = 6).

**Figure 4 insects-16-00984-f004:**
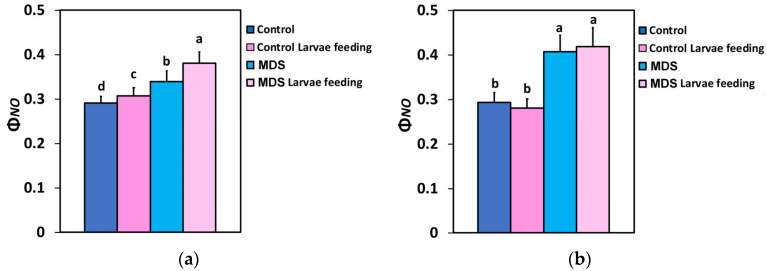
The quantum yield of non-regulated energy loss in PSII (Φ*_NO_*) at 250 μmol photons m^−2^ s^−1^ AL (**a**) and at 1000 μmol photons m^−2^ s^−1^ AL (**b**), before and after 20 min of larvae feeding, in well-watered (WW, control) or mildly drought-stressed (MDS) tomatoes. Standard deviations (SD) are shown by bars. Different lower-case letters reveal significant differences at *p* < 0.05 (*n* = 6).

**Figure 5 insects-16-00984-f005:**
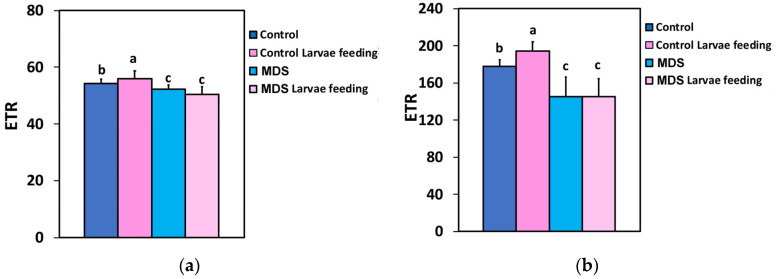
The electron transport rate (ETR) at 250 μmol photons m^−2^ s^−1^ AL (**a**) and at 1000 μmol photons m^−2^ s^−1^ AL (**b**), before and after 20 min of larvae feeding, in well-watered (WW, control) or mildly drought-stressed (MDS) tomatoes. Standard deviations (SD) are shown by bars. Different lower-case letters reveal significant differences at *p* < 0.05 (*n* = 6).

**Figure 6 insects-16-00984-f006:**
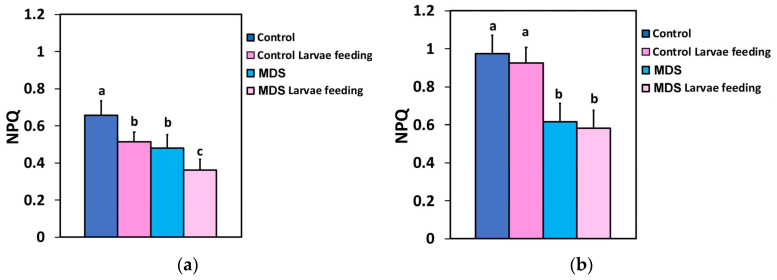
The non-photochemical quenching (NPQ) at 250 μmol photons m^−2^ s^−1^ AL (**a**) and at 1000 μmol photons m^−2^ s^−1^ AL (**b**), before and after 20 min of larvae feeding, in well-watered (WW, control) or mildly drought-stressed (MDS) tomatoes. Standard deviations (SD) are shown by bars. Different lower-case letters reveal significant differences at *p* < 0.05 (*n* = 6).

**Figure 7 insects-16-00984-f007:**
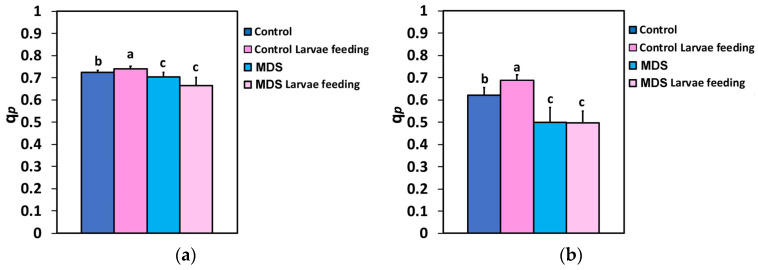
The portion of open PSII rection centers (RCs) (q*p*) that reveal the redox state of quinone A (Q_A_), at 250 μmol photons m^−2^ s^−1^ AL (**a**) and at 1000 μmol photons m^−2^ s^−1^ AL (**b**), before and after 20 min of larvae feeding, in well-watered (WW, control) or mildly drought-stressed (MDS) tomatoes. Standard deviations (SD) are shown by bars. Different lower-case letters reveal significant differences at *p* < 0.05 (*n* = 6).

**Figure 8 insects-16-00984-f008:**
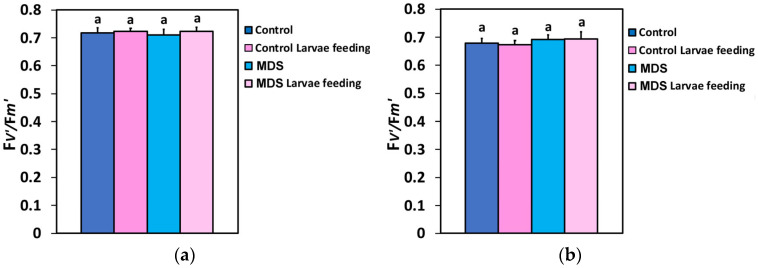
The efficiency of the open PSII RCs (F*v*′/F*m*′) at 250 μmol photons m^−2^ s^−1^ AL (**a**) and at 1000 μmol photons m^−2^ s^−1^ AL (**b**), before and after 20 min of larvae feeding, in well-watered (WW, control) or mildly drought-stressed (MDS) tomatoes. Standard deviations (SD) are shown by bars. Different lower-case letters reveal significant differences at *p* < 0.05 (*n* = 6).

**Figure 9 insects-16-00984-f009:**
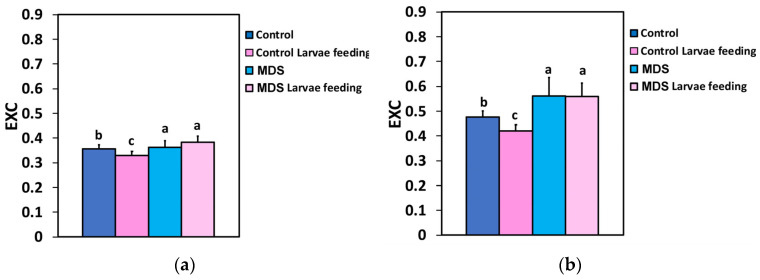
The excess excitation energy at PSII (EXC) at 250 μmol photons m^−2^ s^−1^ AL (**a**) and at 1000 μmol photons m^−2^ s^−1^ AL (**b**), before and after 20 min of larvae feeding, in well-watered (WW, control) or mildly drought-stressed (MDS) tomatoes. Standard deviations (SD) are shown by bars. Different lower-case letters reveal significant differences at *p* < 0.05 (*n* = 6).

**Figure 10 insects-16-00984-f010:**
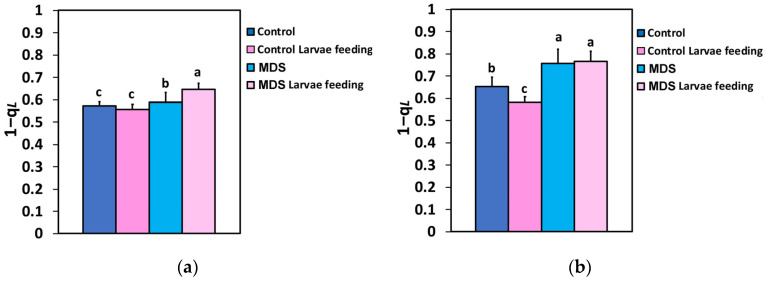
The excitation pressure at PSII (1-q*L*) at 250 μmol photons m^−2^ s^−1^ AL (**a**) and at 1000 μmol photons m^−2^ s^−1^ AL (**b**), before and after 20 min of larvae feeding, in well-watered (WW, control) or mildly drought-stressed (MDS) tomatoes. Standard deviations (SD) are shown by bars. Different lower-case letters reveal significant differences at *p* < 0.05 *(n* = 6).

**Figure 11 insects-16-00984-f011:**
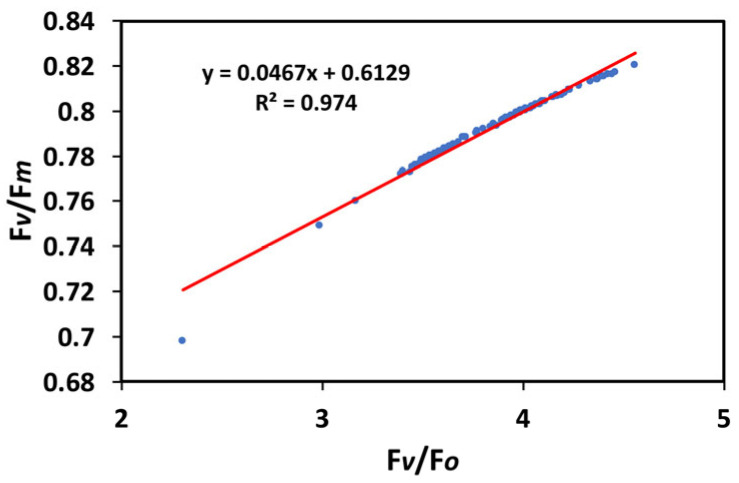
The relationship of the maximum efficiency of PSII photochemistry (F*v/*F*m*) with the efficiency of the oxygen-evolving complex (F*v/*F*o*), before and after 20 min of larvae feeding, in well-watered (WW, control) or mildly drought-stressed (MDS) tomatoes, based on the data of Figure 1a,b. Blue dots represent the paired measurements of the variables while the red line is the regression line that illustrates the relationship between the two variables.

**Figure 12 insects-16-00984-f012:**
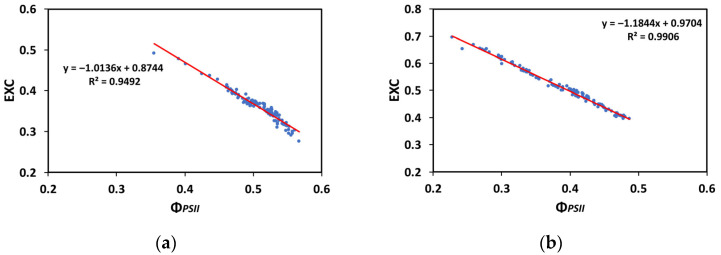
The relationship between the excess excitation energy (EXC) and the effective quantum yield of PSII photochemistry (Φ*_PSII_*) at 250 μmol photons m^−2^ s^−1^ (**a**) and at 1000 μmol photons m^−2^ s^−1^ AL (**b**), before and after 20 min of larvae feeding, in well-watered (control) or mildly drought-stressed tomatoes, based on the data of Figure 2a,b and Figure 9a,b. Blue dots represent the paired measurements of the variables while the red line is the regression line that illustrates the relationship between the two variables.

**Figure 13 insects-16-00984-f013:**
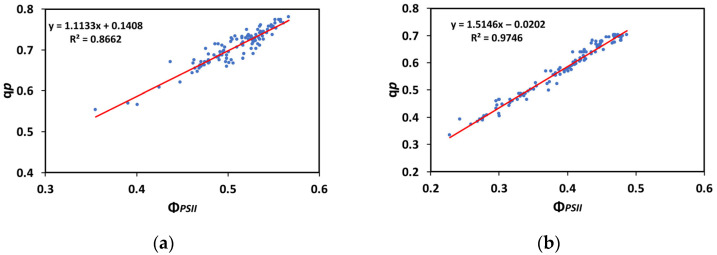
The relationship between the redox state of quinone A (Q_A_) and the effective quantum yield of PSII photochemistry (Φ*_PSII_*) at 250 μmol photons m^−2^ s^−1^ (**a**) and at 1000 μmol photons m^−2^ s^−1^ AL (**b**), before and after 20 min of larvae feeding, in well-watered (control) or mildly drought-stressed tomatoes, based on the data of Figure 2a,b and Figure 7a,b. Blue dots represent the paired measurements of the variables while the red line is the regression line that illustrates the relationship between the two variables.

**Figure 14 insects-16-00984-f014:**
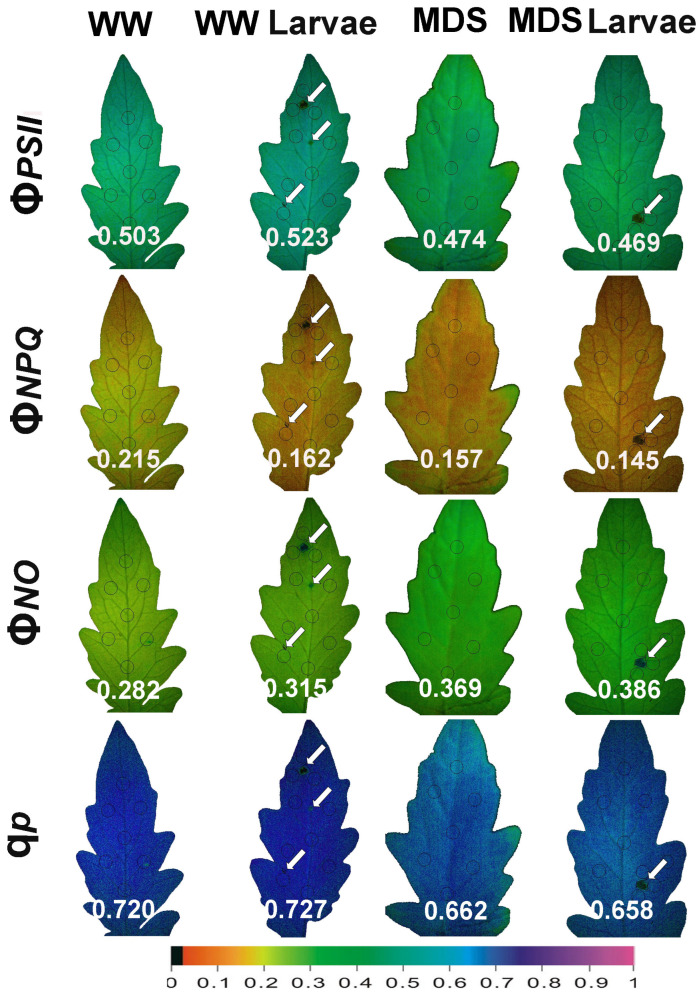
Representative color-coded images of the light energy partitioning at PSII, to photochemistry (Φ*_PSII_*), heat dissipation (Φ*_NPQ_*), or nonregulatory lost (Φ*_NO_*), and the respective color-coded images of the portion of open PSII rection centers (RCs) (q*p*), before and after 20 min of larvae feeding, in well-watered (WW, control) or mildly drought-stressed (MDS) tomatoes. The areas of interest (AOIs) measured at the leaf surface are shown by circles, while the whole leaflet (average) value is given in white. White arrows point out the feeding areas. The color code on the bottom of the images shows pixel values ranging from 0.0 to 1.0.

**Table 1 insects-16-00984-t001:** The leaf water content of well-watered tomato plants and mildly drought-stressed tomatoes and the soil water content of their respective pots (*n* = 10 ± SD).

Parameter	Well-Watered Plants	Mildly Drought-Stressed
Leaf Water Content ^1^	86.19 ± 0.97	83.99 ± 0.08
Soil Water Content ^2^	0.51 ± 0.03	0.22 ± 0.05

^1^ expressed %; ^2^ expressed in m^3^ m^−3^.

## Data Availability

All data supporting the findings of this study are available within the paper and within its Appendix A published online.

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
