# Peer review of "The Compensatory Response of Photosystem II Photochemistry to Short-Term Insect Herbivory Is Suppressed Under Water Deficit"

_insects, 2025, doi:10.3390/insects16090984_

Round 1

Reviewer 1 Report

Comments and Suggestions for Authors

This is a fine study that documents the interrelationships between moderate drought stress and herbivory on photochemistry. The experimental design is sound, the analysis is appropriate, and the results are presented in clear figures that aid the reader in understanding plant responses. I think the authors' conclusions are sound and well supported by the evidence presented. On balance, I have no substantive criticisms. I did find a few areas where the language could be improved, which are:

line 71 "...resulting in..." rather than "..resulting to..."

line 78 change "Drought stress is decreasing photosynthetic efficiency and is changing the biosynthesis..." to "Drought stress decreases photosynthetic efficiency and changes the biosynthesis..."

line 120-122 As written this is quite awkward. I suggest changing it to "...limited investigations have been conducted on the photosynthetic response to insect herbivore feeding..."

Author Response

This is a fine study that documents the interrelationships between moderate drought stress and herbivory on photochemistry. The experimental design is sound, the analysis is appropriate, and the results are presented in clear figures that aid the reader in understanding plant responses. I think the authors' conclusions are sound and well supported by the evidence presented. On balance, I have no substantive criticisms. I did find a few areas where the language could be improved, which are:

Thank you for your positive comments and the English language corrections.

line 71 "...resulting in..." rather than "..resulting to..."

Answer: We changed it.

line 78 change "Drought stress is decreasing photosynthetic efficiency and is changing the biosynthesis..." to "Drought stress decreases photosynthetic efficiency and changes the biosynthesis..."

Answer: We changed the sentence as you suggested.

line 120-122 As written this is quite awkward. I suggest changing it to "...limited investigations have been conducted on the photosynthetic response to insect herbivore feeding..."

Answer: We changed the sentence to “limited investigations have been contacted on how the photosynthetic mechanism responds to insect herbivore feeding”.

Reviewer 2 Report

Comments and Suggestions for Authors

Comments for authors

Title

The title is informative but wordy. Consider rephrasing as “The Compensatory Response of Photosystem II Photochemistry to Short-Term Insect Herbivory is Suppressed under Water Deficit.”

Abstract

The abstract clearly summarizes findings but is overly dense. Some sentences are long and difficult to follow. For example: “The effective quantum yield of PSII photochemistry (ΦPSII), and respectively the electron transport rate (ETR), of the whole leaf…”

Rephrase sentence “overcompensated for the decreased ΦPSII…” to “compensated for the reduction in ΦPSII…”

Keywords

tomato leafminer duplicates Tuta absoluta; keep the scientific name and add the host plant instead (e.g., Solanum lycopersicum / “tomato”) for better retrieval.

drought stress / water deficit, herbivory, non-photochemical quenching (NPQ), oxygen-evolving complex (OEC), excitation pressure, compensatory photosynthesis / hormesis, all are discussed in Results/Discussion. Adding 2–4 lines of these will strengthen discoverability.

crop damage, climate change, quantum yield: replace with field-standard, specific terms used in the paper (e.g., drought stress / water deficit, non-photochemical quenching (NPQ), ΦPSII / effective quantum yield of PSII, compensatory photosynthesis / hormesis).

Introduction

Several sentences are long and contain minor grammatical errors e.g., “limited investigation has been contacted on how does photosynthetic mechanism response to insect herbivore feeding”, should be: “limited investigation has been conducted on how the photosynthetic mechanism responds to insect herbivore feeding.”

Overuse of “due to” (could alternate with “because of,” “resulting from”).

Suggest sharper statement of research gap at the end of the introduction. Currently it is implied but not explicitly highlighted.

Materials and methods

Clarify how many replicates were used for fluorescence imaging and statistical tests. Some sections only mention n=10 for water content but not for fluorescence.

“Shapiro-Wilk test” spelling: remove extra space in “Shapiro -Wilk test” 

Results

In section 3.1, the observed small reduction in leaf water content compared to soil water content is interesting but not fully highlighted in the results. This deserves emphasis, as it indicates potential buffering capacity of the leaf tissue.

In table 1, decimal places are inconsistent (e.g., 0.968 vs 0.078). Standardize rounding across all values preferably two decimal places for uniformity.

There is frequent repetition of what is already stated in figure legends. The text should focus on interpretation and trends, while the legends provide descriptive details.

In results, subheadings are overly long, keep them precise and concise.

Some parts read more like Discussion (e.g., mentioning “overcompensation” or “hormetic responses”). These should be shifted to the Discussion section. The Results should remain descriptive and objective.

Where the results show compensation under WW but suppression under MDS, explicitly summarize direction of effect rather than repeating figure values.

Discussion

I would suggest structuring into sub-themes: (a) compensatory PSII response to herbivory under well-watered conditions, (b) suppression of compensation under mild drought, (c) possible mechanisms (hormesis, ROS, NPQ regulation), (d) ecological/agronomic implications.

The phrase “The effectiveness of hormesis largely depends on dose range, exposure duration, and experimental setup” is accurate but too generic; relate specifically to 20 min feeding used here and how a longer duration might change responses.

Some sections read like a review article rather than specific discussion of current findings. The authors should return frequently to their own data to anchor interpretations.

The discussion misses opportunities to link to field-level implications: if drought suppresses compensatory PSII response, does this translate to yield loss or higher susceptibility to insect pests in tomato crops? This applied context would enhance impact.

The ecological implication that combined stresses will be “inevitable under climate change” is valid, but avoid sweeping statements such as “must be expected.” Rephrase to “is increasingly likely.

Conclusion

The statement that combined stresses “must be expected in future climate conditions” is too strong and speculative. It would be better phrased as “are increasingly likely under projected climate change scenarios.”

The applied significance is underdeveloped. How might these findings affect crop productivity, stress management strategies, or breeding programs for stress resilience? A concluding statement on this would greatly enhance the relevance.

The conclusion could also highlight limitations (e.g., short-term insect feeding, focus only on PSII, lack of biochemical validation of hormesis) and propose future research directions, which would strengthen the scientific balance.

References

The reference list requires careful revision for formatting consistency. For example, entries such as [19], [43], [48], and [56] do not follow the same style as the rest of the references.

Author Response

Thank you for your critical comments that helped us to improve our manuscript. The manuscript was revised considering your comments.

Title

The title is informative but wordy. Consider rephrasing as “The Compensatory Response of Photosystem II Photochemistry to Short-Term Insect Herbivory is Suppressed under Water Deficit.”

Answer: We changed the title following your suggestion

Abstract

The abstract clearly summarizes findings but is overly dense. Some sentences are long and difficult to follow. For example: “The effective quantum yield of PSII photochemistry (ΦPSII), and respectively the electron transport rate (ETR), of the whole leaf…”

Answer: The sentence was shortened.

Rephrase sentence “overcompensated for the decreased ΦPSII…” to “compensated for the reduction in ΦPSII…”

Answer: The sentence “overcompensated for the decreased ΦPSII” was changed to “compensated for the reduction in ΦPSII”

Keywords

tomato leafminer duplicates Tuta absoluta; keep the scientific name and add the host plant instead (e.g., Solanum lycopersicum / “tomato”) for better retrieval.

drought stress / water deficit, herbivory, non-photochemical quenching (NPQ), oxygen-evolving complex (OEC), excitation pressure, compensatory photosynthesis / hormesis, all are discussed in Results/Discussion. Adding 2–4 lines of these will strengthen discoverability.

crop damage, climate change, quantum yield: replace with field-standard, specific terms used in the paper (e.g., drought stress / water deficit, non-photochemical quenching (NPQ), ΦPSII / effective quantum yield of PSII, compensatory photosynthesis / hormesis).

Answer: The key words were changed following your suggestions

Introduction

Several sentences are long and contain minor grammatical errors e.g., “limited investigation has been contacted on how does photosynthetic mechanism response to insect herbivore feeding”, should be: “limited investigation has been conducted on how the photosynthetic mechanism responds to insect herbivore feeding.”

Answer: We changed the sentence to “limited investigations have been contacted on how the photosynthetic mechanism responds to insect herbivore feeding”.

Overuse of “due to” (could alternate with “because of,” “resulting from”).

Answer: Yes, there was overuse of “due to” but now there are only two “due to” left on the whole manuscript.

Suggest sharper statement of research gap at the end of the introduction. Currently it is implied but not explicitly highlighted.

Answer: A sharper statement of the research gap was added.

Materials and methods

Clarify how many replicates were used for fluorescence imaging and statistical tests. Some sections only mention n=10 for water content but not for fluorescence.

Answer: We inserted in all chlorophyll fluorescence figures (n=6).

“Shapiro-Wilk test” spelling: remove extra space in “Shapiro -Wilk test”

Answer: Ok corrected.

Results

In section 3.1, the observed small reduction in leaf water content compared to soil water content is interesting but not fully highlighted in the results. This deserves emphasis, as it indicates potential buffering capacity of the leaf tissue.

Answer: Section 3.1 was rewritten and a paragraph was included in Discussion section highlighting this.

In table 1, decimal places are inconsistent (e.g., 0.968 vs 0.078). Standardize rounding across all values preferably two decimal places for uniformity.

Answer: We standardized bot leaf water content and soil water content values in two decimal places.

There is frequent repetition of what is already stated in figure legends. The text should focus on interpretation and trends, while the legends provide descriptive details.

Answer: Some changes were done in Results section.

In results, subheadings are overly long, keep them precise and concise.

Answer: All subheadings in Results section were shortened.

Some parts read more like Discussion (e.g., mentioning “overcompensation” or “hormetic responses”). These should be shifted to the Discussion section. The Results should remain descriptive and objective.

Answer: There are no mentioning of “overcompensation” or “hormetic responses” in Results section

Where the results show compensation under WW but suppression under MDS, explicitly summarize direction of effect rather than repeating figure values.

Answer: Some changes were done in Results section.

Discussion

I would suggest structuring into sub-themes: (a) compensatory PSII response to herbivory under well-watered conditions, (b) suppression of compensation under mild drought, (c) possible mechanisms (hormesis, ROS, NPQ regulation), (d) ecological/agronomic implications.

Answer: The suggested sub-heading structure of Discussion and especially sub-theme (c) on possible mechanisms (hormesis, ROS, NPQ regulation) is connected to both sub-themes (a) and (b). Thus, it will be an overlapping.

The phrase “The effectiveness of hormesis largely depends on dose range, exposure duration, and experimental setup” is accurate but too generic; relate specifically to 20 min feeding used here and how a longer duration might change responses.

Answer: The paragraph was extended including discussion referring to 20 min and longer duration feeding.

Some sections read like a review article rather than specific discussion of current findings. The authors should return frequently to their own data to anchor interpretations.

Answer: We connected general discussion statements with our results.

The discussion misses opportunities to link to field-level implications: if drought suppresses compensatory PSII response, does this translate to yield loss or higher susceptibility to insect pests in tomato crops? This applied context would enhance impact.

Answer: In conclusion section we mention “Plant health status” seems to be critical in the response of PSII photochemistry to insect herbivory. Since drought stress suppresses PSII compensatory response to insect herbivory, it seems that the increased duration, frequency, and intensity of drought stress episodes owing to climate change, will likely result to higher yield loss attributed to the higher susceptibility of tomatoes to insect pests.”

The ecological implication that combined stresses will be “inevitable under climate change” is valid, but avoid sweeping statements such as “must be expected.” Rephrase to “is increasingly likely.

Answer: This phrase was changed

Conclusion

The statement that combined stresses “must be expected in future climate conditions” is too strong and speculative. It would be better phrased as “are increasingly likely under projected climate change scenarios.”

Answer: The statement was rewritten.

The applied significance is underdeveloped. How might these findings affect crop productivity, stress management strategies, or breeding programs for stress resilience? A concluding statement on this would greatly enhance the relevance.

Answer: Such a concluding statement was included.

The conclusion could also highlight limitations (e.g., short-term insect feeding, focus only on PSII, lack of biochemical validation of hormesis) and propose future research directions, which would strengthen the scientific balance.

Answer: Future research directions were also included.

References

The reference list requires careful revision for formatting consistency. For example, entries such as [19], [43], [48], and [56] do not follow the same style as the rest of the references.

Answer: These references are not articles and require different style according to journal Instructions.

Round 2

Reviewer 2 Report

Comments and Suggestions for Authors

No further comments.